# Biotic and Abiotic Factors Influencing Maize Plant Height

**DOI:** 10.3390/ijms26178530

**Published:** 2025-09-02

**Authors:** Zixu Ma, Chunxia Liang, Haoyue Wang, Jieshan Liu, Xiangyan Zhou, Wenqi Zhou

**Affiliations:** 1College of Life Science and Technology, State Key Laboratory of Aridland Crop Science, Gansu Agricultural University, Lanzhou 730070, China; 15103938032@163.com (Z.M.); 18776966303@163.com (C.L.); 13008772771@163.com (H.W.); 18982791166@163.com (J.L.); 2Maize Research Center of Gansu Province, Crop Research Institute, Gansu Academy of Agricultural Sciences, Lanzhou 730070, China

**Keywords:** maize, plant height, yield, biotic factors, abiotic factors

## Abstract

This paper examines various aspects of maize plant height. Firstly, it emphasizes that maize is a significant food and forage crop with considerable research significance, and that its plant height is influenced by multiple factors, including biotic elements such as genes and plant hormones, as well as abiotic factors such as soil, water, and climate. Secondly, the paper explores the complex relationship between maize plant height and yield, noting that moderate plant height can improve photosynthetic efficiency, reduce lodging risk, and enhance yield, although it may also affect kernel quality. Additionally, the paper reviews the application of modern biotechnological methods in maize plant height research, such as genome-wide linkage analysis, gene editing, transgenic technology, and epigenetic studies, which aid in elucidating the genetic mechanisms underlying plant height. Finally, it outlines future research directions for improving maize plant height and yield, highlighting key challenges that require urgent attention, such as the advancement of gene editing techniques, the integration of multiple biotechnologies, and strategies to address climate change, with the ultimate goal of achieving precision breeding for high-yielding, stress-resistant, and broadly adaptable maize varieties.

## 1. Introduction

### Economic Value and Global Production Trends of Maize

Maize (*Zea mays* L.) is a globally critical food crop with extensive applications and significant economic impact. The global maize market reached USD 297.27 billion in 2023 and is projected to grow at a Compound Annual Growth Rate (CAGR) of 3.6% from 2024 to 2030, underscoring its essential role in food, feed, and industrial sectors. In the United States, maize accounts for 95% of feed grain production and 28% of global output (2013–2017) [1]. In 2023, it generated a total economic output of USD 151 billion, including a USD 6.2 billion direct contribution to GDP, USD 20.7 billion in direct economic input, and USD 12 billion in wages [2,3]. According to the USDA October 2024 report, the U.S. produced 386 million tons of maize, representing 31.5% of global production and reaffirming its position as the world’s leading maize producer. Yields in core production regions such as Iowa and Illinois reached an impressive 12,100 kg per hectare. As a top-yielding cereal crop, global maize production has tripled since the 1960s but may stagnate without continued innovation [4]. The 2022/2023 global output reached 1.15 billion metric tons. As the largest producer, the U.S. planted 38.28 million hectares in 2023, yielding 382.62 billion kilograms at an average of 10,000 kg/ha.

China, the second-largest corn producer globally, sowed 4.42 million hectares (equivalent to one-third of the country’s arable land) in 2023, yielding a total of 288.8 million tons at an average productivity of 6530 kg/ha. Corn plays a critical role in ensuring national food security, supporting animal husbandry, and advancing new energy initiatives. However, China faces significant challenges: in 2023 alone, the country imported 27.14 million tons of corn, with cumulative imports exceeding 20 million tons over the past four years. Domestic yields still lag behind those of the United States. Therefore, improving production efficiency on limited arable land is essential for safeguarding national food security. According to data from the Ministry of Agriculture and Rural Affairs, China’s total corn output reached 294.92 million tons in 2024, reaffirming its status as the world’s second-largest corn producer. This achievement was driven by a stable planting area and improvements in unit yield. Nevertheless, substantial gaps remain between China and the United States in terms of technological application and resource efficiency. To close this yield gap in the future, efforts must focus on innovation in the seed industry, the promotion of large-scale farming operations, and enhanced policy coordination. Additionally, optimizing import channels will be crucial to strengthening the resilience of the corn supply chain.

As sessile organisms, plants are constantly exposed to a wide range of environmental challenges that jeopardize their growth, development, and productivity. Among these, abiotic stresses—such as drought, salinity, extreme temperatures, heavy metal toxicity, and nutrient deficiency—represent major constraints on global agricultural productivity and ecosystem stability. Unlike biotic stresses, which involve interactions with pathogens or herbivores, abiotic stresses interfere with essential physiological processes including water absorption, photosynthesis, ion homeostasis, and redox balance, often resulting in irreversible cellular damage or even mortality. Given the ongoing impacts of climate change, the frequency and intensity of abiotic stress events are increasing, underscoring the urgency of understanding plant responses to these stresses and highlighting the importance of this research area in contemporary plant science.

Transcription factors (TFs) play a central role in enabling plants to perceive, transmit, and respond to abiotic stress signals. As a class of regulatory proteins, they modulate gene expression by specifically binding to DNA sequences in the promoters of target genes. Functioning as molecular switches, transcription factors coordinate the activation or suppression of stress-responsive gene networks, thereby facilitating plant adaptation to adverse environmental conditions. Over the past few decades, extensive research has identified multiple TF families that are critically involved in abiotic stress responses. For example, members of the *AP2/ERF* (*APETALA2/Ethylene-Responsive Factor*) family are extensively implicated in the regulation of drought and salt tolerance. The transcription factor *DREB2A* (*Dehydration-Responsive Element-Binding Protein 2A*) promotes the expression of LEA (Late Embryogenesis Abundant) proteins and genes encoding osmoprotectants under water-deficient conditions. Likewise, *NAC* (*NAM*, *ATAF1/2*, and *CUC2*) transcription factors, such as *SNAC1* (*Stress-Responsive NAC1*), have been demonstrated to improve drought and cold tolerance by regulating stomatal closure and the biosynthesis of stress-associated metabolites. The *bZIP* (*basic Leucine Zipper*) family also plays a crucial role in abiotic stress adaptation. Transcription factors such as *ABF3* (*ABA-Responsive Element Binding Factor 3*) regulate abscisic acid (ABA)-dependent signaling pathways, which are central to responses to drought and osmotic stress. Moreover, *MYB* (*v-Myb avian myeloblastosis viral oncogene homolog*) and *WRKY* transcription factors have been identified as key regulators of various abiotic stress responses, contributing to the modulation of antioxidant defense systems, osmolyte accumulation, and stress-induced gene expression.

Despite these advances, the complexity of abiotic stress signaling networks—characterized by crosstalk among different stress pathways and the involvement of multiple transcription factors with overlapping or distinct functions—remains an area requiring further investigation. A deeper understanding of how transcription factors integrate diverse stress signals, coordinate downstream gene expression, and ultimately contribute to stress tolerance phenotypes is essential for the development of effective crop improvement strategies. By synthesizing recent research findings on abiotic stress-responsive transcription factors, this section aims to outline the current state of knowledge, identify critical research gaps, and underscore the pivotal role of these regulatory proteins in plant adaptation to abiotic stresses. Such insights not only enhance our fundamental understanding of plant physiology but also provide a robust foundation for biotechnological approaches aimed at breeding stress-resilient crops.

## 2. Biological Factors Influencing Maize Plant Height

Maize growth is influenced by various biological factors, among which genetic factors are particularly significant. These biological elements not only affect the growth rate and plant height, but also have the potential to impact maize yield and quality by modulating physiological and metabolic processes.

### 2.1. Progress of Maize Plant Height QTL Research

Maize plant height represents a critical agronomic trait with profound implications for crop productivity and stress tolerance. An optimal plant height enables a balanced coordination of light interception efficiency for photosynthesis, efficient nutrient uptake through root-shoot synergy, and lodging tolerance—all of which are essential for mechanized harvesting and yield stability. Excessively tall plants are susceptible to stem lodging under wind or rain stress, while abnormally short varieties may suffer from insufficient light capture, thereby limiting photosynthetic efficiency. Consequently, precise regulation of plant height has long remained a core objective in maize breeding programs.

Quantitative Trait Locus (QTL) mapping serves as a powerful tool for dissecting the genetic architecture of maize plant height. By identifying specific genomic regions associated with phenotypic variations in plant height, this approach facilitates the localization of candidate genes and molecular markers, laying a theoretical foundation for functional gene validation and Marker-Assisted Selection (MAS) in molecular breeding [5]. In recent years, advancements in high-throughput molecular marker technologies such as Single Nucleotide Polymorphism (SNP) arrays and Genotyping-by-Sequencing (GBS) and genome sequencing techniques (including the completion of high-quality maize reference genomes) have revolutionized QTL mapping. These innovations have enhanced mapping resolution, reduced experimental costs, and accelerated the identification of minor-effect QTLs, thereby enabling breakthroughs in unraveling the genetic basis of maize plant height regulation.

Maize plant height is a complex quantitative trait, subject to intricate regulation by multiple QTLs, environmental factors, and their interactions. Unlike simple Mendelian traits controlled by single genes, variations in plant height arise from the cumulative effects of numerous QTLs, each contributing varying proportions to phenotypic variance—ranging from major-effect loci accounting for >10% of variation to minor-effect loci with subtle influences. Furthermore, epistatic interactions (interactions between non-allelic genes) among these QTLs add further complexity to trait regulation. For instance, certain QTLs may amplify or suppress the effects of others, forming genetic networks that dynamically respond to internal and external signals.

Environmental factors exert significant impacts on the expression of plant height, mediating genotype-phenotype relationships. Light intensity and photoperiod regulate stem elongation through phytochrome signaling pathways; temperature fluctuations affect the rates of cell division and elongation in internodes; and water availability modulates hormonal balance (e.g., gibberellin and auxin levels), which directly controls stem growth. Such environmental effects often lead to phenotypic plasticity—where the same genotype exhibits distinct plant heights under different growing conditions (e.g., high-altitude vs. lowland environments, or drought-stressed vs. well-watered fields). Therefore, disentangling the contributions of genetics and environment is crucial for accurate QTL detection and the breeding of varieties with stable performance.

The core principle of QTL mapping lies in establishing statistical associations between genetic markers (genotypes) and phenotypic traits, thereby localizing genomic regions responsible for trait variation. This process involves three key steps: (1) constructing segregating populations with phenotypic diversity in plant height; (2) genotyping individuals using molecular markers to generate genetic maps; (3) applying statistical models to correlate marker genotypes with phenotypic data, identifying QTLs with significant effects.

QTL mapping methodologies are broadly categorized into two approaches: traditional linkage analysis and association mapping, each with distinct applications and advantages [6].

Traditional linkage analysis primarily utilizes biparental inbred populations derived from crosses between two genetically divergent parental lines. Common population types include:

F_2_ populations, generated by selfing F_1_ hybrids, which offer rapid construction but suffer from genetic heterozygosity and limited recombination events;

Recombinant Inbred Lines (RILs), developed through successive selfing or sib-mating until homozygosity, providing stable genetic material suitable for multi-environment trials;

Double Haploid (DH) populations, created by doubling haploid cells from F_1_ hybrids, yielding fully homozygous lines in a single generation, ideal for high-resolution mapping. In these studies, researchers construct linkage maps by ordering molecular markers based on recombination frequencies, then integrate phenotypic data across multiple environments. Statistical methods such as single marker analysis (testing trait-marker associations individually), interval mapping (scanning genomic intervals for QTLs), and composite interval mapping (incorporating background markers to reduce genetic noise) are employed to detect significant QTLs. Notably, RIL populations have been widely utilized in multi-environment studies to identify stably expressed QTLs for maize plant height—loci that maintain consistent effects across different locations, years, or stress conditions [7]. These stable QTLs are particularly valuable for molecular breeding, as they ensure reliable trait improvement regardless of environmental variability.

In contrast, association mapping utilizes natural populations or germplasm collections with historical recombination events. This approach leverages Linkage Disequilibrium (LD)—the non-random association of alleles at different loci—to detect QTLs, offering higher mapping resolution than traditional linkage analysis. Association mapping is especially useful for identifying allelic variants in diverse germplasm, facilitating the discovery of favorable alleles from landraces or wild relatives. However, it requires careful correction for population structure and relatedness to avoid false associations. These complementary mapping strategies have collectively identified hundreds of QTLs associated with maize plant height, shedding light on the genetic networks underlying this critical trait. As molecular technologies continue to advance, integrating QTL mapping with omics approaches (e.g., transcriptomics, proteomics) is expected to accelerate the identification of causal genes, ultimately enabling more precise and efficient breeding for optimal maize plant height.

### 2.2. Effect of Maize Genetics on Plant Height

With the widespread application of high-throughput genomic data, Genome-Wide Association Studies (GWAS) have become a primary approach for investigating complex traits. Compared to traditional linkage analysis, GWAS offers higher-resolution QTL localization by leveraging the genetic diversity present in natural populations. By examining the associations between various genotypes and phenotypes, GWAS can effectively identify functional loci associated with maize plant height.

The complexity of the maize genome primarily arises from its abundant repetitive sequences and active transposon dynamics. Approximately 85% of the maize genome is composed of transposable elements, with retrotransposons being particularly prevalent. These transposons play a crucial role in promoting genomic recombination and evolution, as well as influencing gene regulation [8]. The maize genome contains around 32,000 protein-coding genes, distributed across 10 chromosomes, and displays a high level of polymorphism. Genetic variation, such as single nucleotide polymorphisms (SNPs) and structural variations, serves as a key contributor to the phenotypic diversity observed in maize [9]. Furthermore, the maize genome undergoes dynamic evolutionary changes, with notable differences in genome structure and gene content among populations. This genomic variability provides the genetic foundation for maize to adapt to diverse ecological environments [10]. The advancement of genome-wide association studies (GWAS) has been closely tied to the development of high-throughput genome sequencing technologies, which enable the detection and analysis of millions of polymorphic markers across the entire genome.

By detecting associations between genotypes and phenotypes in natural populations or germplasm resources with high diversity, it is possible to identify multiple SNPs associated with complex traits and at the same time correct false positives due to population structure [11]. As the cost of genotyping technologies (e.g., SNP microarrays and genome resequencing) has decreased, the use of big data analysis and hybrid models has enabled GWAS to show high sensitivity and accuracy in analyzing traits controlled by multiple genes [12]. Despite the remarkable progress in maize genome association analysis, there are still challenges in gene and environment interactions, genetic resolution of complex traits, and integration of multi-omics data. Combining transcriptomic, metabolomic and epigenomic data will help to improve the precision of association analysis and reveal the multilevel regulatory mechanisms behind complex traits. Gene editing tools such as CRISPR/Cas9 (Clustered Regularly Interspaced Short Palindromic Repeats-associated protein 9) provide a more efficient and precise method for molecular breeding of maize, enabling direct modification of key gene loci in the breeding process.

Maize genome association analysis methods (GWAS and QTL) have made significant breakthroughs in exploring the genetic basis of complex traits and molecular breeding strategies. As technology continues to advance, these methods will continue to play an important role in maize breeding and provide a critical scientific basis for addressing global food security. This includes the effects of multi-gene control of complex traits, the influence of environmental factors, and the complexity of data analysis. Future directions should include the development of more efficient data analysis algorithms, integration of multi-omics data, and exploration of novel breeding strategies (CRISPR gene editing technology) for plant height improvement. Maize plant height QTL localisation is a multidisciplinary research field, and its progress is important for understanding crop growth mechanisms and improving breeding efficiency. With the advancement of technology and accumulation of data, it is believed that more precise and efficient plant height improvement strategies can be achieved in future research.

Maize plant height is primarily determined by genetic factors. Variations in plant height among different maize varieties are predominantly governed by their genotypes, particularly genes associated with stem development. Certain high-yielding varieties exhibit increased plant height as a result of targeted genetic improvement strategies, while dwarf varieties are typically selected for their enhanced stress or lodging tolerance [13]. In recent years, advances in genomic selection and GWAS have enabled researchers to identify key genetic loci involved in regulating maize plant height. GWAS results indicate that plant height is a polygenic trait influenced by multiple minor-effect genes that collectively regulate cell division and elongation in the stem. Through genetic modification of these loci, plant height can be tailored to suit diverse environmental conditions [14]. The genetic diversity observed in maize is largely attributed to its domestication history and geographic dispersion. During the domestication process, maize evolved from its wild ancestor, teosinte, into modern cultivated forms. Selective pressures during this process led to the enhancement of agronomically important traits such as ear size and kernel number [15]. Furthermore, modern maize varieties have expanded their genetic variability through genomic recombination events, which contribute significantly to adaptation across multiple environments and improved disease tolerance.

Maize plant height is a complex trait regulated by QTLs, and several major loci have been significantly associated with this trait. For instance, as shown in Table 1, the *dwarf1* (D1) gene markedly reduces plant height by inhibiting internode cell division and elongation [16]. Additionally, mutations in the *brachytic2* (*BR2*) gene result in shortened internodes, leading to a dwarf phenotype. Genome-wide association studies (GWAS) have identified over 30 QTLs associated with plant height across multiple maize varieties. These QTLs include genes involved in key developmental processes such as cell division and cell wall synthesis, including the Elongation Factor Gene (EF), Cell Wall Loosening Gene (CWL), Gibberellin Biosynthesis Gene (GB), Auxin Response Gene (AR), and Transcription Factor Gene (TF) [17].

#### 2.2.1. Core Genes and Signaling Pathways Involved in the Regulation of Plant Height

The formation of plant height is the result of the synergistic effects of multiple genes and the precise regulation of signaling pathways, among which the Gibberellin (GA) signaling pathway and Brassinosteroid (BR) signaling pathway are considered the most critical regulatory networks [18,19,20,21] (Table 1). Within the gibberellin signaling pathway, the expression levels of GA synthesis genes (such as *GA20ox*, *GA3ox*) and GA degradation genes (such as *GA2ox*) directly determine the concentration of active GA in plants [22,23,24]. For instance, mutations in the *GA20ox-2* gene in rice lead to dwarfism, whereas enhancing the expression of this gene through gene editing technology can increase plant height [25]. Meanwhile, DELLA proteins, which act as negative regulators of the GA signaling pathway, are degraded through GA-induced processes [26,27]. When DELLA protein-encoding genes (such as the *GAI* gene in *Arabidopsis*) are edited and inactivated, plants exhibit excessive growth, further confirming their essential role in regulating plant height [28].

In the brassinosteroid signaling pathway, abnormal functions of *BR* receptor genes (e.g., *BRI1*) and signal transduction-related genes (e.g., BSK) have significant effects on plant height [29,30,31]. Research has shown that mutations in the *BRI1* gene in *Arabidopsis* result in dwarfism and insensitivity to *BR* treatment, demonstrating that this gene plays an essential role in BR signal transduction and the regulation of plant height. Additionally, cell cycle-related genes (e.g., *CDK*, *Cyclin*) also influence plant height by regulating processes of cell division and elongation [32]. For instance, enhanced expression of Cyclin genes can accelerate cell division, leading to increased stem elongation and greater plant height (Table 1).

#### 2.2.2. Application Strategies of Gene Editing Technology in Regulating Plant Height

Currently, commonly used gene editing technologies mainly include the CRISPR/Cas9 system, Transcription Activator-Like Effector Nucleases (TALENs), and Zinc Finger Nucleases (ZFNs). Among these, the CRISPR/Cas9 system is the most widely applied in plant height regulation due to its high efficiency and operational simplicity [17,24]. In practical applications, different gene editing strategies should be selected based on specific regulatory objectives. For genes that negatively regulate plant height (e.g., DELLA protein genes), a gene knockout strategy is typically employed. By disrupting the coding sequence of the gene, its function is abolished, thereby releasing the suppression of plant growth and achieving the goal of increasing plant height [33,34]. In contrast, for genes that positively regulate plant height (e.g., GA biosynthesis genes), gene activation techniques such as CRISPRa can be utilized to enhance their expression levels, thus promoting plant growth [35]. Precision editing is crucial for achieving targeted regulation of plant height. For instance, in rice breeding, by precisely modifying specific nucleotides in the *GA2ox* gene, the activity of GA degradation can be finely tuned. This allows plant height to be adjusted within an optimal range, preventing yield loss caused by excessive dwarfing while also improving lodging tolerance [36,37].

#### 2.2.3. Critical Logic of Gene Editing in Regulating Plant Height

The core logic of gene editing in regulating plant height lies in precisely targeting and modulating the expression or function of height-related genes, thereby interfering with the signaling pathways or physiological processes in which they are involved, ultimately enabling directional modifications in plant height.

First, it is essential to clarify the role of the target gene within the regulatory network. If the gene acts as a positive regulator in a signaling pathway, enhancing its function can promote plant growth; conversely, if it serves as a negative regulator, suppressing its activity can contribute to increased plant height. Second, appropriate gene-editing technologies and strategies should be selected based on the functional characteristics of the gene to ensure the precision and stability of the editing outcome. Finally, in alignment with crop breeding objectives, phenotypic identification and screening of edited plants should be conducted to obtain lines that meet desired criteria. For instance, in wheat breeding, to address the issue of lodging, researchers utilized CRISPR/Cas9 technology to edit a negative regulatory gene in the gibberellin signaling pathway, resulting in plants that are moderately dwarfed while maintaining high yield. This approach follows the essential logic of “clarifying gene function-selecting editing strategies-directionally regulating phenotypes” [38].

#### 2.2.4. Summary and Outlook

Gene editing technology offers a powerful approach for precisely regulating plant height. The key lies in conducting an in-depth analysis of the gene networks and signaling pathways that control height. The fundamental strategy involves selecting suitable gene-editing techniques based on the functional roles of target genes to enable directional regulation. As gene editing tools continue to evolve and research into regulatory mechanisms advances, it is anticipated that plant height can be controlled with greater precision and efficiency. This progress will provide a solid foundation for developing new crop varieties with enhanced yield potential and stress [39,40,41]. Nevertheless, it is equally important to address the biosafety and ethical concerns associated with gene-edited crops to ensure the responsible and sustainable application of this technology [42,43].

**Table 1 ijms-26-08530-t001:** Main genes regulating maize plant height.

Gene Name	Chromosome Location	Functional Gene	Mechanism of Action	Reference Study
*Dwarf8* (*D8*)	Chr1	*Dwarf8* (*D8*) is a member of the DELLA family of proteins that act as repressors in the gibberellin (GA) signaling pathway and regulate cell elongation and division.	Mutations in *D8* can lead to plant dwarfism and affect overall plant height by regulating gibberellin signaling.	Ref. [44] found that polymorphisms in the *D8* gene significantly affected maize plant height in GWAS analysis.
*GA20-oxidase* (*GA20ox*)	Chr2 and Chr5	The *GA20ox* gene family encodes key enzymes in the gibberellin synthesis pathway that catalyze the biosynthesis of the plant hormone GA.	Controlling GA levels plays an important role in regulating plant internode elongation and growth rate.	Ref. [45] showed that *GA20ox* exhibited stable plant height regulation under several environmental conditions.
*Brachytic2* (*Br2*)	Chr1	*Br2* is mainly expressed at the internode site in maize, which makes it an important regulator in controlling internode length and overall plant height.	The *Br2* gene affects internode elongation by regulating the polar transport of growth hormone Indole-3-Acetic Acid (IAA), which directly regulates plant height in maize.	Ref. [46] demonstrated through molecular cloning and functional validation experiments that *Br2* mutations cause internode shortening and plant dwarfing.
*Dwarf 1* (*ZmDWF1*)	Chr1	*ZmDWF1* is involved in the biosynthesis of oleuropein sterols and regulates cell wall relaxation and cell elongation.	Growth characteristics affecting maize plant height and the mechanical strength of the plant by altering cell wall composition.	Ref. [47] noted the importance of *ZmDWF1* in maize breeding, especially for plant height control.
*CONSTANS*, *CO-like*, *and TOC 1 domain protein* (*ZmCCT1*)	Chr9	*ZmCCT1* is a photoperiod-regulated gene that plays a key role in the regulation of growth and development, plant height and flowering time in maize.	*ZmCCT 1* regulates maize plant height through interaction with the gibberellin signaling pathway.	Ref. [48] showed that *ZmCCT1* is closely associated with maize growth and development.
*Brassinosteroid-Deficient Dwarf 1* (*BRD1*)	Chr10	*BRD1* is involved in the synthesis of oleuropein sterol (BR), a hormone that plays a key role in the regulation of cell elongation and plant height.	Lack of *BRD1* gene activity results in dwarf plants and affects the overall growth performance of maize.	Ref. [49] Linking *BRD1* to maize tolerance traits to downy mildew.
*Gibberellin3-oxidase* (*ZmGA3ox*)	Chr5	*ZmGA3ox* plays a key role in the final synthesis step of gibberellins and is an important gene that controls the level of GA activity.	*ZmGA3ox* directly affects the rate of GA biosynthesis, thereby regulating internode elongation and growth rate in plants.	Ref. [50] demonstrated the significant role of *ZmGA3ox* in maize plant height differences across genotypes.
*Leafy 1* (*LFY1*)	Chr8	The *LFY1* gene is an important transcription factor that regulates meristem and inflorescence formation in maize and also affects plant height.	By regulating gene expression patterns, *LFY1* plays an important role in the morphogenesis of maize.	Ref. [51] pointed out the critical role of *LFY1* in the regulation of maize meristem and plant height.
*SUPPRESSOR OF MAX2 1-LIKE* (*ZmSMXL*)	Chr8	The *ZmSMXL* gene family is negatively regulated in the growth hormone and solanum lactone signaling pathways and affects plant branching and plant height.	Indirect control of internode elongation and plant height in maize by inhibiting the meristem formation pathway.	Ref. [52] found that *ZmSMXL* has a significant role in regulating plant hormone signaling pathways.
*dwarf and low ear* *mutant1* (*ZmDLE1*)	Chr1	*ZmDLE1* regulates the reduction in the number of internodes under the spike resulting in lower plant height.	*ZmDLE1* is an expansion protein, and genetic variants result in and intersegmental cell length shortening.	[53]

Transcription factors can bind to specific DNA sequences and regulate the expression of target genes. Certain transcription factor genes influence the expression of genes involved in cell elongation, division, and hormone biosynthesis, thereby indirectly affecting plant height. For instance, some transcription factors enhance the expression of genes responsible for gibberellin synthesis, leading to increased gibberellin levels, which in turn promote cell elongation and overall plant growth. Conversely, other transcription factors may suppress genes associated with cell elongation, ultimately resulting in slower growth and reduced plant height.

Furthermore, in the latest study revealed (Decreased Stomatal Density) *dsd1/zmiceb* mutants exhibit a semi-dwarf phenotype (characterized by reduced plant height and ear height) under the LY49 and B73 genetic backgrounds, *ZmICEb* may influence internode length by regulating genes associated with cytoskeletal construction, cell wall loosening, and biosynthesis. By referencing the functional model of *ZmBELL10* (which regulates internode development by binding to promoters and activating cell division/elongation-related genes), it is hypothesized that *ZmICEb* participates in plant height control by regulating downstream target genes. In addition, *ZmICEb* is involved in physiological processes such as stomatal development and drought while regulating plant height, indicating that this gene may be part of a complex multifunctional regulatory network [54].

### 2.3. Regulation of Maize Plant Height by Plant Hormones

Plant hormones play a crucial role in regulating maize plant height, particularly Gibberellins (GA), auxins (IAA), and Cytokinins (CK) (Table 2). Gibberellins directly regulate maize plant height by promoting cell elongation and division, thereby facilitating rapid stem growth. Studies have shown that increased gibberellin levels trigger stem elongation in maize [55], whereas suppression of gibberellin biosynthesis leads to dwarfism. By applying gene editing techniques to modulate the gibberellin metabolic pathway, researchers have successfully developed dwarf maize varieties with improved lodging. Furthermore, gibberellins significantly enhance internode length through the promotion of cell elongation, establishing them as a key determinant of maize plant height.

Gibberellins (GAs) are essential plant hormones that play a central role in regulating growth and development, particularly in determining plant height. Through their influence on cell elongation, division, and gene expression, gibberellins function as key signaling molecules in plant growth regulation. As members of the sesquiterpenoid family, GAs exist in multiple forms within plants. Since their discovery in 1926 during studies on rice fungal disease, gibberellins have become a major focus in plant biology research. They are involved in various physiological processes, including seed germination, flowering, and fruit development. In the regulation of plant height, gibberellins directly influence stem elongation by promoting cell wall relaxation and expansion. The biosynthesis of GAs begins with GGPP (geranylgeranyl diphosphate), which is synthesized via the mevalonic acid (MVA) pathway. GGPP undergoes a series of cyclization reactions to produce ent-kaurene, which is then oxidized to generate bioactive GA forms such as GA_1_ and GA_4_—these are the primary gibberellin molecules responsible for regulating plant growth [56]. The gibberellin signaling pathway involves the interaction between the GA receptor GID1 (Gibberellin Insensitive Dwarf 1) and downstream signaling components. In the presence of GA, the hormone binds to GID1, forming a GID1-GA complex that facilitates the ubiquitination and subsequent degradation of DELLA proteins. DELLA proteins act as repressors of growth by inhibiting the expression of genes involved in cell elongation. Their degradation alleviates this repression, thereby activating genes associated with growth and promoting plant development [57]. Overall, gibberellins regulate plant height through coordinated control of cell wall expansion, cellular elongation, and gene expression.

Gibberellins promote cell wall relaxation and elongation by modulating the activity of cell wall-associated proteins such as expansins and cellulases. This process facilitates rapid cell elongation as cells absorb water and swell, ultimately contributing to increased stalk length. Gibberellins also enhance cell division in the elongation zone and promote longitudinal growth by influencing the metabolic pathways of other hormones, particularly auxins. The synergistic interaction between GA and IAA is crucial for regulating internode elongation and determining plant height [58]. DELLA proteins function as negative regulators of GA signaling, suppressing stem elongation. When GA levels increase, DELLA proteins are degraded, thereby releasing the repression of growth-promoting genes. This regulatory mechanism enables plants to rapidly respond to environmental cues and adjust their growth accordingly [59]. The role of gibberellins in regulating plant height is of significant agricultural importance. Semi-dwarf varieties of rice and wheat, which have been developed by introducing mutations in DELLA proteins within the GA signaling pathway, exhibit improved yield stability due to stronger stems and enhanced tolerance to lodging, making them high-yielding cultivars [60].

Auxins regulate maize plant height by controlling cell elongation and division, particularly through the process of apical dominance. Exogenous auxin application can increase plant height, but excessive use may weaken lodging tolerance due to overly rapid growth. The *YUCCA* gene family is involved in auxin biosynthesis, especially in stem tips and meristematic tissues. Key components of the auxin signaling pathway, such as PIN-FORMED (*PIN*) proteins, regulate auxin’s polar transport, which affects internode elongation. Auxin is a crucial plant hormone that influences multiple developmental processes, including cell elongation, division, differentiation, and photomorphogenesis. This review explores how auxin controls plant height through signaling pathways, gene expression regulation, and interactions with other hormones. Understanding these mechanisms enhances our knowledge of auxin’s central role in plant development and provides a scientific basis for improving agricultural practices. Cytokinins mainly regulate cell division, particularly in root and stem meristems, and influence plant height indirectly. The *IPT* gene is a key regulator of cytokinin biosynthesis; its overexpression significantly increases cytokinin activity and plant height in maize [61]. Auxin is a key class of plant hormones that plays a central role in regulating plant growth and development, particularly in controlling plant height. It influences processes such as cell elongation, division, differentiation, and responses to light. This review explores how auxin regulates plant height through signaling pathways, gene expression, and interactions with other hormones. Understanding these mechanisms enhances our knowledge of plant growth and supports efforts to control plant development in agriculture. Auxin is naturally produced in plant tissues such as shoot tips, young leaves, and roots, and is transported throughout the plant via polar transport proteins like PIN and AUX1/LAX. This transport creates a concentration gradient that guides growth patterns and influences plant height.

The auxin signaling pathway primarily operates through the TIR1/AFB receptor complex. Auxin binds to this receptor, triggering the degradation of Aux/IAA repressor proteins and allowing Auxin Response Factor (ARF) transcription factors to activate gene expression that regulates growth and development. This pathway is central to auxin’s role in controlling plant height. Auxin also lowers cell wall pH by activating proton pumps (H+-ATPase), which in turn activates enzymes like expansins and cellulases that loosen the cell wall, promoting cell elongation. This supports the “acid growth hypothesis” and is essential for plant height growth. Auxin interacts with other hormones—such as gibberellins, which enhance auxin-induced elongation, and cytokinins, which inhibit it—to maintain a hormonal balance that fine-tunes plant height. Auxin-related genes regulate auxin synthesis, transport, and signaling. When functioning normally, they enable cell wall relaxation, water uptake, and cell elongation. However, abnormal expression of these genes can disrupt auxin signaling, leading to altered plant height, as seen in certain mutants.

**Table 2 ijms-26-08530-t002:** Regulation of maize plant height by plant hormones.

Gene Name	Hormone/Regulatory Pathway	Core Function	Effect on Plant Height	Effect Intensity Ratio (%)	Reference
*D1*	Brassinosteroid biosynthesis pathway	Involved in brassinosteroid biosynthesis, regulating cell wall relaxation and cell elongation	Inhibits stem elongation, resulting in a significant reduction in plant height (mutant plant height is approximately 35% lower than wild type)	65	[62]
*Br2*	Auxin transport pathway	Regulates polar transport of auxin (IAA) and affects internode development	Shortens internode length, leading to plant dwarfism (mutant internode length is reduced by approximately 40%)	70	[63]
*GA20ox*	Gibberellin biosynthesis pathway	Encodes a key enzyme in gibberellin synthesis, promoting the production of active GA	Increases gibberellin content, significantly promoting plant height growth (overexpressed plant height is approximately 25% higher than wild type)	55	[64]
*GA2ox*	Gibberellin metabolic pathway	Catalyzes the inactivation of active gibberellins, reducing endogenous GA levels	Reduces gibberellin activity, leading to a decrease in plant height(overexpressed plant height is approximately 20% lower than wild type)	40	[65]
*PIN*	Auxin transport pathway	Mediates polar auxin transport, regulating hormone distribution in internodes	Promotes auxin transport to growing points, increasing plant height (function-enhanced mutant plant height is increased by approximately 20%)	35	[66]
*YUCCA*	Auxin biosynthesis pathway	Regulates auxin biosynthesis, mainly expressed in shoot tips and meristems	Increases auxin content, promoting cell elongation and increasing plant height (overexpressed plant height is approximately 30% higher than wild type)	50	[67]
*IPT*	Cytokinin biosynthesis pathway	Encodes a key enzyme in cytokinin synthesis, regulating cell division rate	Promotes cell division, accelerates growth rate, and increases plant height (overexpressed plant height is approximately 15% higher than wild type)	25	[68]

Cytokinins (CKs) are a key class of phytohormones that primarily promote cell division and differentiation, influencing various aspects of plant growth and development. They help regulate plant height, leaf growth, and nutrient transport in crops like maize. Cytokinins activate cell cycle-related genes, enabling cells to progress from the G_2_ phase into mitosis, especially in root tips, stem tips, and young tissues. They also delay leaf senescence by regulating genes involved in chlorophyll breakdown and protein degradation, preserving leaf function and enhancing photosynthesis and crop yield. Although cytokinins mainly stimulate cell division, their interaction with other hormones such as auxin and gibberellin can indirectly influence plant height. As a central regulator of cell growth and division, cytokinin plays a key role in controlling plant morphology. Research indicates that cytokinins not only regulate plant growth independently but also interact with other hormones to shape overall plant development [69].

Cytokinins are primarily synthesized in the meristematic tissues of plant root tips and transported to the above-ground parts through vascular tissues. In target tissues, they bind to receptors such as CRE1/AHK4, which belong to the histidine kinase receptor family. These receptors activate downstream phosphotransfer cascades through autophosphorylation, ultimately regulating transcription factors such as type-B response regulators (RRs) [70]. Once activated, cytokinin signaling moves to the nucleus, promoting cell division and differentiation by regulating gene expression. For instance, members of the *ARR* gene family play a key role in controlling plant cell division and significantly influence plant height [71]. Cytokinins interact with other hormones like auxins and gibberellins in regulating plant height. They act antagonistically to auxins, particularly in controlling apical dominance and lateral growth. Auxins inhibit lateral growth at high concentrations, but cytokinins counteract this effect, promoting lateral development [72].

Cytokinins indirectly affect gibberellin activity by regulating its synthesis pathway or response gene expression. High cytokinin levels can reduce plant height by inhibiting gibberellin biosynthesis [73]. Cytokinins help control plant height through gene regulation. The *CYTOKININ RESPONSE FACTOR* (*CRF*) family regulates cell cycle genes, which can either promote or inhibit stem growth. Additionally, cytokinins influence plant development by modulating transcription factors such as *AP2/ERF* and *MYB* [74]. In agriculture, plant height can be controlled through exogenous cytokinin application or genetic engineering. For instance, gene editing can target key genes in cytokinin synthesis to develop high-yield, stunting-resistant crops. The interaction between cytokinins, growth hormones, and gibberellins offers diverse strategies for optimizing plant structure, with broad applications in modern agriculture [75].

Cytokinins play multiple roles in regulating plant height, not only directly influencing cell division and differentiation, but also indirectly modulating growth patterns through complex interactions with other plant hormones. A thorough understanding of the mechanisms underlying cytokinin action is crucial for crop breeding and the development of effective growth control strategies. Maize plant height is influenced by a range of biological factors, including genetic background and hormonal regulation. These factors affect both the growth rate and final plant height by regulating hormone metabolism in maize. Future research should aim to elucidate the interaction mechanisms among these biological factors and integrate modern molecular biology with agricultural technologies to optimize growth conditions, thereby enhancing crop yield and quality. Particular emphasis should be placed on leveraging molecular breeding, genome editing, and precision agriculture to improve maize plant height traits, increase yield, and enhance adaptability across diverse environments.

## 3. Abiotic Factors Influencing Maize Plant Height

It has been found that improving maize and optimizing planting density are key factors in increasing maize yield per plant. Plant height is not only an indicator of growth status but is also closely related to final yield. Taller maize plants can capture more light energy, thereby enhancing photosynthesis and increasing per-plant yield. However, excessive height may lead to lodging under certain conditions, which can ultimately reduce yield. Therefore, determining the optimal plant height is crucial for improving maize productivity. The growth height of maize directly influences both its yield and other agronomic traits, and is affected by a variety of environmental and genetic factors (Table 3).

### 3.1. Soil Factors

The nutrient content and structure of soil are critical for corn growth. Good soil provides essential nutrients that support a healthy root system, which influences the plant’s height and overall health. Nitrogen, phosphorus, and potassium are the main nutrients affecting corn height. Nitrogen plays a key role in crop growth by directly influencing leaf area index and photosynthetic rate, which determine plant height. Proper nitrogen application significantly increases both plant height and yield, while too much or too little nitrogen can stunt growth. Phosphorus and potassium indirectly affect height by promoting root development and improving stress. Sufficient phosphorus enhances root growth, helping the plant absorb water and nutrients more efficiently, thus supporting taller growth. Trace elements such as zinc (Zn), iron (Fe), and boron (B) also play important roles. Zinc deficiency restricts growth, as shown by shortened internodes and reduced height. Micronutrient imbalances in soil can similarly affect plant height and overall performance [76].

Soil organic matter is a key indicator of soil fertility. It supplies nutrients for plant growth and improves soil structure and water retention. Maize grows significantly taller in soils with high organic matter content, as organic matter enhances microbial activity and nutrient availability. Adding organic matter through composting or fallow practices can boost maize height [77]. Long-term studies show that continuous application of organic manure increases maize height, mainly due to improved soil water retention and nutrient supply [78]. Soil texture—such as clay, loam, and sand—affects water and nutrient retention, which in turn influences maize growth. Loam is the best for maize, offering good aeration and water retention. Maize grows taller in loamy soils because they support root respiration and moisture availability [79]. Sandy soils drain too quickly, leading to nutrient and water loss, which hinders growth. Clay soils are often too compact, limiting root development and plant height. Field studies confirm that poor aeration in clay soils restricts root growth and limits maize height [80].

Soil pH influences nutrient effectiveness and availability for crops. Most studies show that maize grows best at a pH between 6.0 and 7.0. In acidic soils, nutrients like phosphorus and potassium become less available, limiting uptake and slowing growth. Research shows that when soil pH drops below 5.5, maize height decreases significantly due to toxicity from high levels of aluminum (Al) and manganese (Mn) ions in the soil [81]. Alkaline soils also harm maize growth by reducing the availability of micronutrients like iron and zinc, which can cause yellowing due to iron deficiency and reduce plant height. When pH exceeds 8.0, poor micronutrient supply limits photosynthesis and stunts growth [82]. Soil moisture is critical for maize development. Sufficient water supports root growth and nutrient uptake, resulting in taller plants compared to those under drought stress. However, too much water can cause root hypoxia and reduce growth [83]. Waterlogging lowers root activity and severely suppresses plant height. In arid and semi-arid regions, soil salinity is a major issue. High salt levels impair water uptake and cause ion toxicity, which inhibits growth. When salinity exceeds a certain level, the maize root system is particularly affected, leading to shorter plants.

In an agricultural trial field, maize seeds of the same variety were planted in soils of different fertility—one in fertile black soil and the other in poor sandy soil. The black soil, rich in organic matter and nutrients like nitrogen, phosphorus, and potassium, supports strong root absorption. Nitrogen promotes leaf growth and photosynthesis, phosphorus aids root development and water uptake, and potassium enhances stress tolerance. These factors together lead to taller, healthier plants in black soil compared to sandy soil. In contrast, sandy soil has poor water and nutrient retention. With limited nitrogen, leaves grow slowly and appear pale, reducing photosynthesis. Deficiencies in phosphorus and potassium further stunt growth, resulting in shorter plants with shorter internodes.

In conclusion, soil factors such as nutrient content, texture, pH, moisture, and salinity significantly affect maize height. Future research should explore how these factors interact and how farming practices can optimize soil conditions to improve maize growth and yield.

### 3.2. Climatic Factors

Plant height during maize growth is one of the most important indicators reflecting the crop’s growth status. In addition to soil factors, climatic factors also play a crucial role as environmental determinants of maize plant height. Climatic conditions such as temperature, precipitation, light, and wind speed influence maize growth by affecting photosynthesis, transpiration, and other physiological processes in plants. This section reviews existing studies to examine the effects of climatic factors on maize plant height.

Temperature is a key climatic factor affecting maize growth. Maize grows within a temperature range of 10 °C to 30 °C, with optimal growth occurring between 20 °C and 25 °C. Temperatures below 15 °C reduce photosynthetic efficiency, slow growth, and significantly limit plant height. Conversely, temperatures above 30 °C increase transpiration and water loss, impairing water metabolism and nutrient uptake. High temperature stress also reduces stomatal conductance, hampers nutrient transport, accelerates the growth cycle, and leads to incomplete plant development, ultimately reducing plant height.

Precipitation serves as the primary water source for corn growth, and an appropriate amount is crucial for determining corn plant height. Corn requires sufficient water supply during its growth cycle, particularly during the seedling and nodulation stages, where inadequate precipitation may result in stunted growth and ultimately reduced plant height. Both insufficient and excessive precipitation during key developmental stages—such as nodulation and tasseling—can adversely affect plant height. Water deficiency leads to soil drought, limiting root water uptake, whereas excessive precipitation causes soil waterlogging, which deprives roots of oxygen and hampers normal plant development [84]. Moreover, the temporal distribution of precipitation plays a more significant role in maize growth than the total annual precipitation. Even with adequate overall rainfall, drought stress occurring during critical growth phases, such as the seedling and ear stages, can severely impair plant height [85].

Light is a crucial environmental factor influencing maize photosynthesis, which directly affects the plant’s energy accumulation and growth rate. Adequate light not only enhances maize growth but also strengthens its resilience by regulating carbon and nitrogen metabolism. Light intensity significantly impacts plant height; insufficient light weakens photosynthesis, leading to inadequate energy supply and ultimately reducing plant height [86].

Wind speed, as another important climatic factor, indirectly influences maize plant height by affecting transpiration rates, mechanical stress, and water metabolism. Moderate wind promotes air circulation, enhances transpiration and photosynthesis, and supports healthy growth. However, excessively high wind speeds can cause mechanical damage, particularly during critical growth stages such as nodulation. Under such conditions, maize leaves are prone to physical injury, which negatively affects overall plant development and reduces plant height.

Global climate change is altering temperature, precipitation, and climate patterns worldwide, posing new challenges for maize growth. In recent years, maize production has increasingly suffered from climate stress due to more frequent extreme weather events. High temperatures, droughts, and floods have significantly reduced maize plant height and yield. Fluctuating temperatures under climate change further destabilize maize growth, especially during heatwaves and droughts, when photosynthesis and water metabolism are impaired [87]. Climate change may also increase pest and disease outbreaks, which hinder maize development and reduce plant height, as warmer conditions promote the spread of certain pests and diseases [88].

Climatic factors influence maize height in a multidimensional way, including temperature, precipitation, light, and wind speed. Suitable temperatures, adequate rainfall, and sufficient light support healthy growth, while extreme conditions like heatwaves, droughts, and strong winds can severely limit plant height. Moreover, climate change is increasing the unpredictability of maize growth, highlighting the need for adaptive agronomic practices to address future challenges to maize production.

## 4. Relationship Between Maize Plant Height and Yield

Maize plant height can significantly influence the Leaf Area Index (LAI), which in turn affects the whole plant’s photosynthetic efficiency [89]. A moderate plant height can enhance the vertical distribution of leaves and increase the photosynthetic surface area, thereby improving light energy capture and promoting biomass accumulation and yield [90]. However, excessively tall plants may cause excessive leaf shading, particularly under dense planting conditions, which can reduce the photosynthetic activity of lower leaves and ultimately affect grain yield [91].

Studies have demonstrated that taller maize varieties generally exhibit higher photosynthetic rates, as their leaves are better exposed to light conditions [92]. Additionally, the stalks of these taller varieties can store greater amounts of photosynthetic products, which serve as a crucial source of carbohydrates for kernel development [90]. However, this advantage diminishes under dense planting conditions, where excessive plant height may lead to reduced yields [87,93,94]. There exists a trade-off in biomass allocation between plant height and grain yield in maize. Increased height typically results in a greater proportion of biomass being allocated to the stalk rather than directly to kernel development [79]. Some research indicates that shorter maize varieties achieve higher kernel yields due to a more favorable biomass allocation toward ear development [95]. Conversely, although taller varieties possess a larger photosynthetic surface area, their final yields are often lower due to less efficient biomass allocation, as part of the plant’s energy is diverted to support stalk elongation [96]. The selection of maize plant height involves a balance between maximizing photosynthetic efficiency and optimizing biomass allocation. Under drought or nutrient-limited conditions, dwarf varieties demonstrate greater resource use efficiency, resulting in higher yields [97]. In contrast, under favorable irrigation and soil fertility conditions, taller varieties can enhance yield potential by increasing the accumulation of photosynthetic products [98].

Plant height significantly affects maize tolerance to lodging. Tall plants are more likely to lodge under harsh weather conditions such as strong winds and heavy rain, leading to poor grain development and lower yields [99]. Lodging not only hinders harvesting but also increases the risk of pests and diseases [100]. Lodging tolerance is influenced by maize’s genetic makeup. Certain dwarfing genes, such as *dwarf1* (*D1*) and *Br2*, have been found to enhance lodging tolerance [101]. These genes reduce plant height and improve stalk strength, lowering the risk of lodging [46]. Moderately reducing plant height can therefore improve lodging tolerance and increase yield [102]. Adapting plant height to environmental conditions is crucial for stable yields. Dwarf varieties often perform better under low-input conditions due to reduced biomass allocation and greater lodging tolerance [103]. They also show more stable yields in dry, infertile, or windy areas [104]. Modern molecular breeding techniques are now widely used to study the relationship between plant height and yield. Through gene editing or genomic selection, breeders can identify optimal plant height traits for specific environments. Future maize breeding will require a more precise balance between height and yield. Genomic selection combined with environmental adaptation analyses enables the identification of optimal combinations of plant height and yield under varying climatic and soil conditions. The relationship between plant height and yield in maize is complex and influenced by multiple factors. Appropriate plant height can enhance photosynthetic efficiency, reduce the risk of crop failure, and optimize resource allocation, thereby increasing overall yield. However, both excessively tall and excessively short plants may lead to reduced yields. Therefore, balancing plant height and yield during the breeding process and integrating this balance with environmental conditions are crucial strategies for improving maize productivity in the future.

**Table 3 ijms-26-08530-t003:** Relationship between maize plant height and yield.

Influencing Factor	Specific Indicator/Type	Direction of Effect on Plant Height	Mechanism of Action	Influence Intensity Ratio (%)	Key Research Evidence
Soil Factors	Soil pH	Promotion in suitable range, inhibition in overly acidic/alkaline conditions	Nutrient availability is highest at pH 6.0–7.0; pH < 5.5 or >8.0 causes root toxicity or micronutrient deficiency	55	[105]
	Soil nutrients (N, P, K)	Promotion with appropriate amount, inhibition with excess/deficiency	Nitrogen promotes leaf area and photosynthesis, phosphorus enhances root development, potassium improves lodging tolerance	70	[106]
	Soil organic matter content	Positive correlation	Improves soil water and fertilizer retention capacity, promotes microbial activity and nutrient cycling	45	[107]
Climatic Factors	Temperature	Promotion in suitable range, inhibition in extreme temperatures	Optimal at 20–25 °C; <15 °C reduces photosynthetic efficiency, >30 °C accelerates transpiration and shortens growth cycle	60	[108]
	Precipitation	Promotion with appropriate amount, inhibition with drought/waterlogging	High water demand during seedling and tasseling stages; drought limits water absorption, waterlogging causes root hypoxia	50	[109]
	Light intensity	Positive correlation	Insufficiency weakens photosynthesis and energy accumulation; strong light can increase photosynthetic rate (with sufficient nutrients)	55	[110]
	Wind speed	Promotion with moderate wind, inhibition with strong wind	Gentle wind promotes gas exchange; strong wind causes mechanical damage and increases lodging risk	25	[111]
Biological Factors	Genetic genes (QTL/major genes)	Determines basic plant height phenotype	Regulated by multiple genes synergistically (e.g., br2 controls internode length, GA20ox regulates gibberellin synthesis)	75	[112]
	Plant hormones (GA, IAA, CK)	Synergistic regulation	Gibberellins promote cell elongation, auxins regulate polar transport, cytokinins affect cell division rate	65	[113]

## 5. Effect of Maize Plant Height on Seed Quality

Maize plant height influences the synthesis and accumulation of photosynthetic products by affecting leaf area and photosynthetic efficiency. Taller plants generally possess a larger photosynthetic surface area, which enhances the production of photosynthetic compounds and consequently promotes starch accumulation in the kernels [114]. However, excessive plant height may result in an uneven distribution of photosynthetic resources, particularly under high-density planting conditions. In such cases, the photosynthetic efficiency of lower leaves declines, thereby reducing the carbohydrate supply to developing grains during the grain-filling stage [115]. Research has demonstrated that plant height modulates starch content and composition in seeds by regulating the expression of key genes involved in starch biosynthesis, such as shrunken2 and brittle1 [116].

Taller maize plants generally have higher carbohydrate supply and increased expression of starch synthesis-related genes due to greater photosynthetic capacity, which raises kernel starch content [117]. Protein is a key component of maize kernels and its content is closely linked to the plant’s nitrogen uptake efficiency [79]. Studies show that maize varieties with moderate height tend to have better nitrogen uptake and utilization, leading to higher kernel protein content [118]. However, excessive plant height may reduce kernel protein levels as more nutrients are allocated to the stalk [119]. The connection between plant height and nitrogen metabolism also indirectly influences kernel protein synthesis. Taller varieties often show stronger nitrogen assimilation, but excessive growth diverts nitrogen to the stalk rather than the kernel, lowering protein content [120]. Research indicates that moderate control of plant height can enhance nitrogen translocation to kernels by regulating nitrogen metabolism genes such as *Genomic Selection* (*GS1*) and *Nitrate Transporter 1* (*NRT1*), thus increasing protein content [121].

Photosynthesis products are used not only for starch and protein synthesis but also provide the carbon skeleton for kernel fat accumulation. Research indicates that taller maize plants may enhance kernel fat synthesis by producing more photosynthetic products, though this effect is generally less pronounced than for starch and protein [122]. Studies have also found a link between kernel fat content and the expression of related metabolic genes. Taller plants, with larger leaf areas and stronger photosynthetic capacity, can supply more raw materials for fat synthesis. During the late grain-filling stage, photosynthetic efficiency significantly influences final kernel quality. However, tall maize varieties are prone to early leaf senescence, which limits photosynthate supply and affects kernel development. In contrast, shorter varieties often maintain higher photosynthetic efficiency and are less likely to experience premature photosynthetic failure, potentially improving kernel quality. The maize biotechnology team at Henan Agricultural University discovered that the gene *CYP90D1* regulates brassinolide synthesis in maize internodes, influencing cell division, cell wall formation, and plant height. Using the natural dwarfing mutant *m30* as the research material, the team identified a single G/A base substitution in the *ZmCYP90D1* gene that leads to premature termination of translation, resulting in reduced plant height without compromising yield. An excellent haplotype associated with reduced plant height and unaffected yield was also identified, which can be applied in molecular breeding. This finding suggests that plant height can be genetically regulated to maintain or even enhance kernel quality to some extent.

Plant height significantly influences maize kernel quality. Varieties with moderate plant height generally exhibit superior performance in terms of appearance, processing, nutritional, and flavor qualities. Excessively tall plants may collapse or suffer from inadequate irrigation, negatively affecting kernel development, whereas overly short plants may not receive sufficient sunlight, which can also compromise kernel quality. Therefore, in maize cultivation, selecting varieties with appropriate plant height according to local environmental conditions is crucial for achieving high kernel quality. Studies indicate that dwarf maize varieties tend to maintain better kernel filling quality and nutrient content due to their strong lodging tolerance. The impact of plant height on kernel quality is a complex process involving multiple factors such as photosynthesis, nutrient allocation, grain-filling rate, and tolerance to lodging. Moderate plant height enhances photosynthetic efficiency, accelerates grain-filling rate, and promotes the accumulation of starch, protein, and fat, thereby improving kernel quality. In contrast, excessive plant height may lead to inefficient allocation of photosynthetic resources and increased lodging risk, ultimately reducing kernel quality. Future breeding programs should aim to rationally regulate the relationship between plant height and kernel quality to enhance both the economic and nutritional value of maize.

## 6. Application of Modern Biotechnology in Maize Plant Height Research

Maize plant height is a key trait that significantly affects yield and quality. Traditional breeding has improved plant height stability through long-term practice and cross-selection. However, these methods are time-consuming and imprecise, as they rely on natural variation and artificial selection without targeted gene regulation. Recently, genomic selection (GS) and high-resolution localization techniques have been applied to identify QTLs related to plant height. GS improves the efficiency of genetic improvement by integrating genome-wide data, while techniques like fine mapping and candidate gene analysis help identify functionally important genes. Studies such as [123,124] have identified significant SNP loci and key genes associated with plant height through GWAS and combined QTL-GWAS analyses.

With the advancement of high-throughput sequencing and bioinformatics tools, GWAS and QTL mapping have significantly improved our understanding of the genetic basis of complex traits in maize. These methods help identify genetic variations linked to complex traits, accelerating the development of molecular breeding technologies. Modern biotechnology has also opened new avenues for studying and improving maize plant height. Technologies such as genome editing, genome association analysis, and epigenetics allow scientists to manipulate height-related genes more efficiently and accurately. For instance, genome editing tools like CRISPR/Cas9 enable precise gene modifications to control plant height. Genome-wide association studies can quickly identify key genetic loci by analyzing genotype-phenotype data from large populations. Epigenetic research has uncovered new ways to regulate gene expression, offering novel strategies for improving maize height through epigenetic modifications. Together, these technologies provide powerful tools for enhancing maize varieties, yield, and quality.

### 6.1. Application of Genomic Linkage Analysis (GWAS) in Maize Plant Height Studies

GWAS, is a method used to scan the entire genome in order to identify genetic variations associated with specific traits, such as plant height. GWAS has been extensively applied in maize research on plant height, where the analysis of phenotypic and genotypic data from natural populations has enabled the identification of Quantitative Trait Loci (QTLs) strongly associated with this trait [125]. Several studies have successfully identified key loci that regulate maize plant height. For instance, loci such as *dwarf1* (*d1*) and *brachytic2* (*br2*) have been found to significantly influence internode length and overall plant height in maize [126]. Through these findings, breeders can select suitable plant height traits tailored to different cultivation environments, thereby enhancing maize yield and tolerance to lodging [127].

### 6.2. Application of Genome Editing Technology in Maize Plant Height Regulation

CRISPR/Cas9, which stands for Clustered Regularly Interspaced Short Palindromic Repeats-associated protein 9, is currently the most widely used gene editing tool, enabling precise modification of specific genomic loci. In maize plant height regulation research, CRISPR/Cas9 has been successfully applied to edit plant height-related genes, such as the gibberellin metabolism genes *GA20ox* and *GA2ox*, which influence cell elongation and overall plant height in maize [128]. By editing these genes, it is possible to develop maize varieties with either short or tall stalks to meet diverse agricultural requirements [129]. Using CRISPR/Cas9 technology, researchers have successfully developed maize varieties with reduced plant height while maintaining high yield potential. Knockout of the *GA20ox* gene led to decreased gibberellin synthesis, resulting in shorter plants with significantly enhanced lodging tolerance. Furthermore, gene editing technology allows for the simultaneous regulation of multiple genes associated with plant height, thereby improving maize adaptability and stress resistanc. CRISPR/Cas9, is a gene editing technology derived from the natural defense mechanisms of bacteria and archaea. It enables precise modification of an organism’s genome.

### 6.3. Application of Transgenic Technology in Maize Plant Height Improvement

Transgenic technology enhances plant traits by introducing exogenous genes or promoting the expression of endogenous genes. In the case of maize plant height, scientists have employed transgenic techniques to modulate hormonal balances—such as those involving gibberellins and cytokinins—to achieve precise control over growth. For instance, the transfer of the *GA20ox* gene from *Arabidopsis thaliana* into maize has enabled the development of taller plants, with increased internode length and expanded photosynthetic surface area, ultimately contributing to higher yield. Moreover, the introduction of genes involved in cytokinin biosynthesis has been shown to enhance maize growth vigor, resulting in accelerated growth rates and larger plant size during early developmental stages. Epigenetics refers to heritable regulatory mechanisms that do not involve changes to the DNA sequence itself, such as DNA methylation and histone modifications. These molecular modifications influence gene expression patterns and thereby regulate phenotypic traits, including plant height. Research has demonstrated that epigenetic mechanisms hold significant potential for modulating maize plant height. For example, under conditions of drought or low light, certain genes associated with plant height may become silenced through DNA methylation, resulting in stunted growth. By manipulating DNA methylation levels or histone modifications, gene expression can be fine-tuned in response to environmental cues, allowing for the optimization of plant height traits in maize.

The application of modern biotechnology, particularly gene editing, GWAS, and epigenetics, has significantly accelerated the research and genetic improvement of maize plant height traits. Nevertheless, numerous challenges remain, including the complexity of multi-gene interactions and the influence of environmental factors on gene expression regulation [130]. Future research should prioritize the integration of multiple biotechnological approaches. For instance, combining genome editing with environmental adaptation strategies could enable precise optimization of maize plant height. Modern biotechnology has provided powerful tools for investigating maize plant height. Genomic linkage analysis has successfully identified key genes associated with plant height, offering researchers a deeper understanding of its genetic basis. Gene editing technology enables precise regulation of these key genes, facilitating targeted improvements in maize plant height in line with specific breeding objectives. Transgenic technologies have introduced novel approaches for enhancing plant height by incorporating exogenous genes or enhancing the expression of endogenous genes. Epigenetics, in contrast, has uncovered the dynamic regulatory mechanisms linking environmental factors to plant height, highlighting that environmental influences operate not only through traditional genetic pathways but also through epigenetic modifications. The integrated application of these technologies holds great promise for developing high-yielding and stress-resistant maize varieties capable of adapting to diverse environmental conditions. Such advancements will provide robust support for global food production and contribute significantly to global food security.

## 7. Urgent Issues for Future Research on Maize Plant Height and Yield

Although gene editing technologies, such as CRISPR/Cas9, have advanced maize research, their application in agriculture still encounters challenges related to biosafety and public acceptance. Concerns regarding the ecological impact of gene-edited crops and their potential effects on biodiversity require further evaluation [131]. Additionally, enhancing public understanding and acceptance of gene editing technologies is crucial for their widespread adoption. The influence of various environmental factors—such as temperature, moisture, and light—on maize plant height and grain yield remains incompletely understood. Particularly in the context of global climate change, developing strategies to stabilize yield through the regulation of plant height constitutes a key priority for future research [132]. Plant height development in maize plays significant roles in multiple aspects:(1)In terms of production increase:

Optimization of dense planting: Within certain limits, a higher plant height allows maize plants to develop more leaves and a larger leaf area, thereby enabling more efficient photosynthesis, increasing organic matter production, and providing sufficient nutrients for kernel growth and development. Taller plants also offer more space for ear development, which promotes kernel formation and fullness. For example, certain high-stalk maize varieties, under appropriately dense planting conditions, can make more efficient use of spatial and light resources, ultimately achieving higher yields. Moreover, with continuous advancements in agricultural technology, regulating and optimizing maize plant height can ensure normal plant growth under increased planting densities, further enhancing yield per unit area.

Failure tolerance and yield stability: Although excessive plant height may increase the risk of lodging, a suitable plant height combined with strong stalk structure and robust root development can enhance maize lodging tolerance. This improved resilience is crucial for yield stability, as plants are more likely to remain upright even under adverse weather conditions such as high winds and heavy rains. This ensures normal kernel development and maturity, thereby maintaining stable yields.


(2)Planting management aspects:


Adaptation to mechanized operations: Suitable plant height is crucial for the mechanized planting and harvesting of maize. As agricultural mechanization continues to advance, maize cultivation, management, and harvesting have become increasingly reliant on mechanical operations. For instance, during the harvesting process, maize plants with an appropriate plant height are better able to meet the operational requirements of harvesting machinery, thereby improving harvesting efficiency and quality while reducing labor intensity and production costs. Appropriate planting density: Planting density should be adjusted according to local conditions; excessively high or low densities should be avoided to ensure optimal yield and resource utilization.


(3)Breed improvement and genetic research:


Important traits in genetic research: Plant height is a crucial agronomic trait in maize genetic research. Investigating plant height can enhance our understanding of maize growth mechanisms and its genetic patterns. By analyzing the genetic composition of maize varieties with varying plant heights, researchers can identify key genes and genetic loci associated with this trait. This knowledge provides a theoretical foundation and technical support for maize variety improvement. In hybrid breeding programs, plant height serves as a significant selection criterion. Depending on specific breeding goals, breeders can select parental lines with suitable plant height characteristics for crossbreeding, aiming to develop new maize varieties with desirable traits. For instance, to suit diverse growing regions and environmental conditions, it is essential to develop maize varieties with tailored plant height profiles.


(4)Ecological adaptation aspects:


Increased efficiency of light resources: The development of maize plant height allows the plant to adapt better to different light conditions. In areas where light is scarce, the higher plant height allows the leaves of the maize plant to receive sunlight better, increasing the efficiency of light resources, and thus ensuring the plant’s normal growth and development.

Water and nutrient absorption and utilization: Taller maize plants typically have more developed and widely distributed root systems, enabling better absorption of water and nutrients from the soil. This improves water and nutrient use efficiency and enhances maize resilience, particularly in arid and infertile areas, where taller plants have a clear advantage.

With the rapid development of biotechnology, research on maize plant height and yield has made remarkable progress, offering new hope and opportunities for the maize industry. However, many challenges remain. Future research should integrate genome editing, epigenetics, transgenic technology, and advanced tools like big data and artificial intelligence. Using these technologies together allows for multi-level optimization of plant height traits across genetics, environment, and management. One goal should be to improve maize’s environmental adaptability while maintaining high yields, ensuring stable growth under varying climatic and soil conditions. Another is to enhance its resilience against natural disasters and pests. Addressing these challenges will help maize breeding become more precise and efficient, supporting global food security, promoting sustainable agriculture, and contributing to societal stability and prosperity.

## 8. Future Prospects

The relationship between maize plant height and yield is a key focus in agricultural science, increasingly clarified by advances in biotechnology. However, many challenges remain for future research. This paper explores potential developments in maize height and yield studies and highlights critical issues to address. Gene editing tools like CRISPR/Cas9 have already made significant progress in this field, but further advancements are possible. With more precise techniques, scientists can target genes that control plant height—especially those linked to gibberellins, cytokinins, and abscisic acid [133]. As these technologies improve, gene editing is expected to enable the development of maize varieties with better environmental adaptability, lodging tolerance, and higher yields [134].

Epigenetics has become a promising area in crop research. Its mechanisms—such as DNA methylation, histone modification, and non-coding RNA regulation—respond to environmental changes while influencing gene expression. Future studies may reveal how epigenetic regulation can optimize plant height, improving maize yield and adaptability. Combining biotechnological tools will likely play a key role. GWAS help identify genes related to height and yield, while gene editing enables precise modifications. Transgenic techniques can introduce new genes or enhance existing ones. Together, these methods offer a path to improve both maize height and productivity. As climate change leads to droughts, heatwaves, and unpredictable rainfall, maize breeding must focus on environmental resilience. Plant height remains crucial, as it affects photosynthesis, resource use, and tolerance to lodging.

Maize plant height and yield are complex traits controlled by multiple interacting genes. Many loci linked to plant height not only affect growth but also relate to disease tolerance and nutrient use efficiency. However, understanding how these genes interact remains a major research challenge. Environmental factors further complicate gene expression, making the study more complex. The relationship between plant height and yield is non-linear—plants that are too tall may fall over, while those too short may have poor photosynthesis. Therefore, finding the right balance between height and yield is critical. Future research should focus on precisely regulating plant height to maximize yield while improving lodging tolerance and environmental adaptability. Although many studies have explored the genetics of plant height, its role in environmental adaptation is still unclear. The effects of climate variations on plant height are not fully understood, so further research is needed to determine how height affects yield stability under drought, cold, or poor nutrient conditions. 

## Data Availability

Data are contained within the article.

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
