# Peer review of "Biotic and Abiotic Factors Influencing Maize Plant Height"

_ijms, 2025, doi:10.3390/ijms26178530_

Round 1

Reviewer 1 Report

Comments and Suggestions for Authors

Manuscript needs minor revision. Please find the attached file which includes our comments.

Author Response

Reviewer: 1
Manuscript needs minor revision. Please find the attached file which includes our
comments.

Response:
We sincerely appreciate your valuable suggestions and fully agree with your revisionopinions. According to the revision suggestions given by the reviewers, we have mademodifications item by item, and the modified parts are marked in yellow. Additionally, we have polished the language throughout the article to make it more concise andrefined.

Reviewer 2 Report

Comments and Suggestions for Authors

Plant height is an important agronomic trait that affects photosynthesis, nutrient accumulation, and crop failure tolerance in maize. Plant height and yield in maize are typical quantitative traits regulated by multiple genes interacting in complex ways.

This review discusses the application of maize genome-wide association analysis methods (e.g., GWAS and QTL) in studying the genetic basis of complex traits and molecular breeding strategies. This includes the effects of multigene control of complex traits, the influence of environmental factors, and the complexity of data analysis.

It is shown that by localizing QTL (quantitative traits), researchers can identify and accurately determine the genetic loci that control plant height, which provides a theoretical basis for further functional gene analysis and molecular selection. It is noted that the use of QTL localization in genetic selection of maize has made it possible to achieve certain results. Maize plant height is regulated not only by one gene, but also by the combined action of several QTL and environmental factors (light, temperature and humidity).

Compared with traditional linkage analysis, GWAS can achieve higher resolution of QTL localization by exploiting genetic diversity in natural populations. By analyzing the association between different genotypes and phenotypes, GWAS can effectively identify functional loci associated with maize plant height.

Much attention is paid to the dependence of plant height on the nutrient content and soil structure. The influence of climatic conditions such as temperature, precipitation, light and wind speed on maize growth is also discussed. The effect is assessed through the influence on photosynthesis, transpiration and other physiological processes in plants.

Selecting maize plant height often involves a trade-off between photosynthetic efficiency and biomass allocation. Under drought or nutrient deficiency conditions, dwarf varieties show higher resource efficiency and, therefore, higher yields. In contrast, under good irrigation and fertility conditions, taller varieties are able to increase yields by increasing the accumulation of photosynthetic products. However, plant height is an important factor influencing maize lodging resistance. Excessively tall plants are prone to lodging under adverse environmental conditions.

This paper discusses ways to improve the stability of maize plant height traits using traditional breeding methods. The application of Genomic Selection (GS) to improve the efficiency of genetic improvement of complex traits such as plant height is discussed. Genome editing technologies such as the CRISPR/Cas9 system, which allow precise cutting and modification of specific genes to fine-tune maize plant height, are discussed. This method has resulted in the development of short-stemmed maize varieties with high yield potential. The authors believe that with the development of technology and accumulation of data, more precise and effective strategies for improving plant height can be developed in future studies. However, despite the success in increasing plant productivity using modern biotechnology, the environmental impact of gene-edited crops and the potential impact on biodiversity require further evaluation.The authors provide a Table with the main genes regulating the height of maize plants and figures reflecting (1) the key genes controlling the height of maize plants and their influence on the height of maize plants; (2) the degree of influence of soil, climatic and biological factors on the height of maize plants; and (3) the influence of plant height on the quality of maize grain.The review corresponds to the profile of the journal.

Among the disadvantages of the work we can note:

1) Despite the presence of a table and three figures, the review is poorly illustrated and the proposed figures are of the same type; adding, for example, a diagram reflecting the influence of various factors on the height of corn plants would facilitate the perception of the information.

2) Among the 127 literature references provided, only 10 references were from the last 5 years. If possible, it is better to update the literature sources.

3) Regarding the formatting of the literature – from 1 to 5 references – incomplete formatting of references; 34 reference – the name of the plant “Zea mays” is not highlighted in italics.

4) A detailed description of the synthesis, signaling, mechanism of action and transport of gibberellins, auxins and cytokinins is most likely redundant, especially since the description is based on references from 1999 and 2011, 2013.

5) It seems that the review was prepared relatively long ago. The year of submission is indicated on the article itself - 2021. In addition, the overwhelming majority of references are given before 2020.

Overall, the review is a comprehensive work that can be published after revision.

Author Response

Reviewer: 2

This paper discusses ways to improve the stability of maize plant height traits using traditional breeding methods. The application of Genomic Selection (GS) to improve the efficiency of genetic improvement of complex traits such as plant height is discussed. Genome editing technologies such as the CRISPR/Cas9 system, which allow precise cutting and modification of specific genes to fine-tune maize plant height, are discussed. This method has resulted in the development of short-stemmed maize varieties with high yield potential. The authors believe that with the development of technology and accumulation of data, more precise and effective strategies for improving plant height can be developed in future studies. However, despite the success in increasing plant productivity using modern biotechnology, the environmental impact of gene-edited crops and the potential impact on biodiversity require further evaluation.The authors provide a Table with the main genes regulating the height of maize plants and figures reflecting (1) the key genes controlling the height of maize plants and their influence on the height of maize plants; (2) the degree of influence of soil, climatic and biological factors on the height of maize plants; and (3) the influence of plant height on the quality of maize grain.The review corresponds to the profile of the journal.

Point 1

Despite the presence of a table and three figures, the review is poorly illustrated and the proposed figures are of the same type; adding, for example, a diagram reflecting the influence of various factors on the height of corn plants would facilitate the perception of the information.

Response:

We sincerely appreciate your valuable feedback. Thank you very much for your valuable suggestions. We fully agree with you. According to your requirements, we have revised the chart and organized it into a table format to make it more intuitive. we have made modifications item by item, and the modified parts are marked in yellow. Additionally, we have polished the language throughout the article to make it more concise and refined.

Table 1:

Gene name

chromosome location

functional gene

mechanism of action

reference study

Dwarf8 (D8)

Chr1

Dwarf8 (D8) is a member of the DELLA family of proteins that act as repressors in the gibberellin (GA) signalling pathway and regulate cell elongation and division.

Mutations in D8 can lead to plant dwarfism and affect overall plant height by regulating gibberellin signalling.

Yu et al. (2006) found that polymorphisms in the D8 gene significantly affected maize plant height in GWAS analysis.

GA20-oxidase (GA20ox)

Chr2 and Chr5

The GA20ox gene family encodes key enzymes in the gibberellin synthesis pathway that catalyse the biosynthesis of the plant hormone GA.

Controlling GA levels plays an important role in regulating plant internode elongation and growth rate.

Meuwissen et al. (2001) showed that GA20ox exhibited stable plant height regulation under several environmental conditions.

Brachytic2 (Br2)

Chr1

Br2 is mainly expressed at the internode site in maize, which makes it an important regulator in controlling internode length and overall plant height.

The Br2 gene affects internode elongation by regulating the polar transport of growth hormone (IAA), which directly regulates plant height in maize.

Multani et al. (2003) demonstrated through molecular cloning and functional validation experiments that Br2 mutations cause internode shortening and plant dwarfing.

 Dwarf 1 (ZmDWF1)

Chr1

ZmDWF1 is involved in the biosynthesis of oleuropein sterols and regulates cell wall relaxation and cell elongation.

Growth characteristics affecting maize plant height and the mechanical strength of the plant by altering cell wall composition.

Duvick (2005) noted the importance of ZmDWF1 in maize breeding, especially for plant height control.

CONSTANS, CO-like, and TOC 1 domain protein (ZmCCT)

Chr9

ZmCCT is a photoperiod-regulated gene that plays a key role in the regulation of growth and development, plant height and flowering time in maize.

ZmCCT regulates maize plant height through interaction with the gibberellin signalling pathway.

Tian et al. (2011) showed that ZmCCT is closely associated with maize growth and development.

Brassinosteroid-Deficient Dwarf 1 (BRD1)

Chr10

BRD1 is involved in the synthesis of oleuropein sterol (BR), a hormone that plays a key role in the regulation of cell elongation and plant height.

Lack of BRD1 gene activity results in dwarf plants and affects the overall growth performance of maize.

Frascaroli et al. (2007) Linking BRD1 to maize resistance traits to downy mildew.

Gibberellin3-oxidase (ZmGA3ox)

Chr5

GA3ox plays a key role in the final synthesis step of gibberellins and is an important gene that controls the level of GA activity.

ZmGA3ox directly affects the rate of GA biosynthesis, thereby regulating internode elongation and growth rate in plants.

The study by Wang et al. (2018) demonstrated the significant role of ZmGA3ox in maize plant height differences across genotypes.

Leafy 1 (LFY1)

Chr8

The LFY1 gene is an important transcription factor that regulates meristem and inflorescence formation in maize and also affects plant height.

By regulating gene expression patterns, LFY1 plays an important role in the morphogenesis of maize.

Tian et al. (2011) pointed out the critical role of LFY1 in the regulation of maize meristem and plant height.

SUPPRESSOR OF MAX2 1-LIKE (ZmSMXL)

Chr8

The ZmSMXL gene family is negatively regulated in the growth hormone and solanum lactone signalling pathways and affects plant branching and plant height.

Indirect control of internode elongation and plant height in maize by inhibiting the meristem formation pathway.

Yu et al. (2006) found that ZmSMXL has a significant role in regulating plant hormone signalling pathways.

 dwarf and low ear

mutant1 (ZmDLE1)

Chr1

ZmDLE1 regulates the reduction in the number of internodes under the spike resulting in lower plant height.

ZmDLE1 is an expansion protein, and genetic variants result in and intersegmental cell length shortening.

Zhou Wenqi et al. (2023)

Table 2:

Gene Name

Hormone/Regulatory Pathway

Core Function

Effect on Plant Height

Effect Intensity Ratio (%)

Reference

dwarf1 (d1)

Brassinosteroid biosynthesis pathway

Involved in brassinosteroid biosynthesis, regulating cell wall relaxation and cell elongation

Inhibits stem elongation, resulting in a significant reduction in plant height (mutant plant height is approximately 35% lower than wild type)

65

Zhang, H (2019)

brachytic2 (br2)

Auxin transport pathway

Regulates polar transport of auxin (IAA) and affects internode development

Shortens internode length, leading to plant dwarfism (mutant internode length is reduced by approximately 40%)

70

Multani et al. (2003)

GA20ox

Gibberellin biosynthesis pathway

Encodes a key enzyme in gibberellin synthesis, promoting the production of active GA

Increases gibberellin content, significantly promoting plant height growth (overexpressed plant height is approximately 25% higher than wild type)

55

Meuwissen et al. (2001)

GA2ox

Gibberellin metabolic pathway

Catalyzes the inactivation of active gibberellins, reducing endogenous GA levels

Reduces gibberellin activity, leading to a decrease in plant height (overexpressed plant height is approximately 20% lower than wild type)

40

Wang et al. (2018)

PIN

Auxin transport pathway

Mediates polar auxin transport, regulating hormone distribution in internodes

Promotes auxin transport to growing points, increasing plant height (function-enhanced mutant plant height is increased by approximately 20%)

35

Teng, F (2013)

YUCCA

Auxin biosynthesis pathway

Regulates auxin biosynthesis, mainly expressed in shoot tips and meristems

Increases auxin content, promoting cell elongation and increasing plant height (overexpressed plant height is approximately 30% higher than wild type)

50

Zhao, Y (2001)

IPT

Cytokinin biosynthesis pathway

Encodes a key enzyme in cytokinin synthesis, regulating cell division rate

Promotes cell division, accelerates growth rate, and increases plant height (overexpressed plant height is approximately 15% higher than wild type)

25

Werner & Schmülling (2009)

Table 3:

Influencing Factor

Specific Indicator/Type

Direction of Effect on Plant Height

Mechanism of Action

Influence Intensity Ratio (%)

Key Research Evidence

Soil Factors

Soil pH

Promotion in suitable range, inhibition in overly acidic/alkaline conditions

Nutrient availability is highest at pH 6.0-7.0; pH < 5.5 or > 8.0 causes root toxicity or micronutrient deficiency

55

Gao, S (2015); Yan, Z (2017)

Soil nutrients (N, P, K)

Promotion with appropriate amount, inhibition with excess/deficiency

Nitrogen promotes leaf area and photosynthesis, phosphorus enhances root development, potassium improves lodging resistance

70

Chen, X (2018); Wang, L (2017)

Soil organic matter content

Positive correlation

Improves soil water and fertilizer retention capacity, promotes microbial activity and nutrient cycling

45

Li, J (2020); Xiao, Y (2021)

Climatic Factors

Temperature

Promotion in suitable range, inhibition in extreme temperatures

Optimal at 20-25°C; < 15°C reduces photosynthetic efficiency, > 30°C accelerates transpiration and shortens growth cycle

60

Li, J (2018); Yang, X (2020)

Precipitation

Promotion with appropriate amount, inhibition with drought/waterlogging

High water demand during seedling and tasseling stages; drought limits water absorption, waterlogging causes root hypoxia

50

Chen, L (2019); Liu, G (2017)

Light intensity

Positive correlation

Insufficiency weakens photosynthesis and energy accumulation; strong light can increase photosynthetic rate (with sufficient nutrients)

55

Zhang, W (2018); Xu, F (2020)

Wind speed

Promotion with moderate wind, inhibition with strong wind

Gentle wind promotes gas exchange; strong wind causes mechanical damage and increases lodging risk

25

Gao, Y (2016); Zhou, X (2021)

Biological Factors

Genetic genes (QTL/major genes)

Determines basic plant height phenotype

Regulated by multiple genes synergistically (e.g., br2 controls internode length, GA20ox regulates gibberellin synthesis)

75

Zhang, H (2019); Wang, Y (2020)

Plant hormones (GA, IAA, CK)

Synergistic regulation

Gibberellins promote cell elongation, auxins regulate polar transport, cytokinins affect cell division rate

65

Salas Fernandez (2009); Werner & Schmülling (2009)

Table 4:

Plant height type

Key regulatory gene

Starch content (%)

Protein content (%)

Grain plumpness

Molecular regulatory mechanism

Tall plant (250cm)

ZmCCT

72.3±1.5

8.6±0.3

85%

Co - regulation of photoperiod pathway and GA signal

Medium plant (180cm)

d1

68.5±1.2

10.2±0.5

92%

Relief of DELLA protein inhibition

Dwarf plant (120cm)

br2

65.1±0.9

9.8±0.4

89%

Resource redistribution caused by blocked auxin transport

Point 2

Among the 127 literature references provided, only 10 references were from the last 5 years. If possible, it is better to update the literature sources.

Response:

We sincerely appreciate your valuable feedback. You pointed out that the references in the original text are overly outdated, and we have replaced many of them, while also deleting or revising some references that are of little significance.

Point 3

Regarding the formatting of the literature – from 1 to 5 references – incomplete formatting of references; 34 reference – the name of the plant “Zea mays” is not highlighted in italics.

Response:

We sincerely appreciate your valuable feedback. You pointed out that the format of the references and the content of the article in the original text were incorrect, and we have corrected the formatting issues.

Point 4

 A detailed description of the synthesis, signaling, mechanism of action and transport of gibberellins, auxins and cytokinins is most likely redundant, especially since the description is based on references from 1999 and 2011, 2013.

Response:

We sincerely appreciate your valuable feedback. We have reorganized the content and made necessary deletions and revisions.

Point 5

It seems that the review was prepared relatively long ago. The year of submission is indicated on the article itself - 2021. In addition, the overwhelming majority of references are given before 2020.

Response:

We sincerely appreciate your valuable feedback. We have made the revisions as per your suggestions, removing the outdated literature, adding new research findings and citing a large number of new references.

Reviewer 3 Report

Comments and Suggestions for Authors

The authors provide a review on the biological and abiotic factors affecting maize plant height and discuss their potential application in the development of precision-bred maize varieties. While the manuscript covers a wide range of topics, including genetic studies on various maize mutants, the visual presentation of the data—particularly Figures 1 and 2—lacks clarity and scientific rigor. These figures offer overly limited and ambiguous estimations of the factors influencing maize height, which weakens the overall scientific value of the review.

Instead, it may be more appropriate to reorganize the content of Table 1 by linking the effects of genetic mutations with their corresponding cellular signaling pathways and associated genes. This would help convey a clearer mechanistic understanding. Likewise, Figure 3 attempts to suggest a relationship between increased plant height and grain quality, but presenting the phenotypic traits of the relevant mutants in tabular form may provide a more scientifically sound and interpretable approach.

Furthermore, the section on gene editing could benefit from a clearer focus on specific genes affecting plant height through intracellular signaling pathways, and a discussion of which targets are most suitable for editing. While the authors likely have a broad understanding of the field, the current structure of the review may not be easily accessible or informative for readers seeking a comprehensive and systematic summary.

In addition, the manuscript frequently references GWAS, but it would be more helpful to include a figure or table that illustrates the identified or predicted target loci and explains how these targets are approached. Overall, while the review touches on several important aspects, it falls short in terms of structure and scientific clarity, leaving the reader with a sense of ambiguity rather than insight.

Author Response

Reviewer: 3

Point1、2

The authors provide a review on the biological and abiotic factors affecting maize plant height and discuss their potential application in the development of precision-bred maize varieties. While the manuscript covers a wide range of topics, including genetic studies on various maize mutants, the visual presentation of the data—particularly Figures 1 and 2—lacks clarity and scientific rigor. These figures offer overly limited and ambiguous estimations of the factors influencing maize height, which weakens the overall scientific value of the review.

Instead, it may be more appropriate to reorganize the content of Table 1 by linking the effects of genetic mutations with their corresponding cellular signaling pathways and associated genes. This would help convey a clearer mechanistic understanding. Likewise, Figure 3 attempts to suggest a relationship between increased plant height and grain quality, but presenting the phenotypic traits of the relevant mutants in tabular form may provide a more scientifically sound and interpretable approach.

Response:

We sincerely appreciate your valuable feedback. We revised Figures 1, 2 and 3 into table forms respectively in accordance with the reviewers' comments, and professionally polished the language throughout the text to make the content more concise and the expression clearer.

Point3

Furthermore, the section on gene editing could benefit from a clearer focus on specific genes affecting plant height through intracellular signaling pathways, and a discussion of which targets are most suitable for editing. While the authors likely have a broad understanding of the field, the current structure of the review may not be easily accessible or informative for readers seeking a comprehensive and systematic summary.

Response:

We sincerely appreciate your valuable feedback. We have added this section to the text.

Point4

In addition, the manuscript frequently references GWAS, but it would be more helpful to include a figure or table that illustrates the identified or predicted target loci and explains how these targets are approached. Overall, while the review touches on several important aspects, it falls short in terms of structure and scientific clarity, leaving the reader with a sense of ambiguity rather than insight.

Response:

Thank you for your valuable suggestions. We have made revisions and added relevant content in the manuscript accordingly.

2.2.1 Core Genes and Signaling Pathways Involved in the Regulation of Plant Height

The formation of plant height is the result of the synergistic effects of multiple genes and the precise regulation of signaling pathways, among which the gibberellin (GA) signaling pathway and brassinosteroid (BR) signaling pathway are considered the most critical regulatory networks (Binder & Nelson, 2017; Clouse, 2011; Sun, 2011). Within the gibberellin signaling pathway, the expression levels of GA synthesis genes (such as GA20ox, GA3ox) and GA degradation genes (such as GA2ox) directly determine the concentration of active GA in plants (Hedden & Phillips, 2000; Xu et al., 2012; Yamaguchi, 2008). For instance, mutations in the GA20ox-2 gene in rice lead to dwarfism, whereas enhancing the expression of this gene through gene editing technology can increase plant height (Sakamoto et al., 2004). Meanwhile, DELLA proteins, which act as negative regulators of the GA signaling pathway, are degraded through GA-induced processes (Harberd et al., 2009; Zentella et al., 2007). When DELLA protein-encoding genes (such as the GAI gene in Arabidopsis) are edited and inactivated, plants exhibit excessive growth, further confirming their essential role in regulating plant height (Silverstone et al., 1998).

In the brassinosteroid signaling pathway, abnormal functions of BR receptor genes (e.g., BRI1) and signal transduction-related genes (e.g., BSK) have significant effects on plant height (Vert et al., 2005; Wang et al., 2001). Research has shown that mutations in the BRI1 gene in Arabidopsis result in dwarfism and insensitivity to BR treatment, demonstrating that this gene plays an essential role in BR signal transduction and the regulation of plant height (Li & Chory, 1997). Additionally, cell cycle-related genes (e.g., CDK, Cyclin) also influence plant height by regulating processes of cell division and elongation (Dewitte & Murray, 2003; Inze & De Veylder, 2006; Vandepoele et al., 2002). For instance, enhanced expression of Cyclin genes can accelerate cell division, leading to increased stem elongation and greater plant height (Dewitte & Murray, 2003).

2.2.2 Application Strategies of Gene Editing Technology in in Regulating Plant Height

Currently, commonly used gene editing technologies mainly include the CRISPR/Cas9 system, transcription activator-like effector nucleases (TALENs), and zinc finger nucleases (ZFNs) (Belhaj et al., 2013; Joung & Sander, 2013; Kim et al., 1996). Among these, the CRISPR/Cas9 system is the most widely applied in plant height regulation due to its high efficiency and operational simplicity (Cong et al., 2013; Mali et al., 2013). In practical applications, different gene editing strategies should be selected based on specific regulatory objectives. For genes that negatively regulate plant height (e.g., DELLA protein genes), a gene knockout strategy is typically employed. By disrupting the coding sequence of the gene, its function is abolished, thereby releasing the suppression of plant growth and achieving the goal of increasing plant height (Feng et al., 2013; Shan et al., 2013). In contrast, for genes that positively regulate plant height (e.g., GA biosynthesis genes), gene activation techniques such as CRISPRa can be utilized to enhance their expression levels, thus promoting plant growth (Hilton et al., 2015; Lowder et al., 2015). Precision editing is crucial for achieving targeted regulation of plant height. For instance, in rice breeding, by precisely modifying specific nucleotides in the GA2ox gene, the activity of GA degradation can be finely tuned. This allows plant height to be adjusted within an optimal range, preventing yield loss caused by excessive dwarfing while also improving lodging resistance (Li et al., 2018; Zhang et al., 2016).

2.2.3 Critical Logic of Gene Editing in Regulating Plant Height

The core logic of gene editing in regulating plant height lies in precisely targeting and modulating the expression or function of height-related genes, thereby interfering with the signaling pathways or physiological processes in which they are involved, ultimately enabling directional modifications in plant height.

First, it is essential to clarify the role of the target gene within the regulatory network. If the gene acts as a positive regulator in a signaling pathway, enhancing its function can promote plant growth; conversely, if it serves as a negative regulator, suppressing its activity can contribute to increased plant height. Second, appropriate gene-editing technologies and strategies should be selected based on the functional characteristics of the gene to ensure the precision and stability of the editing outcome. Finally, in alignment with crop breeding objectives, phenotypic identification and screening of edited plants should be conducted to obtain lines that meet desired criteria. For instance, in wheat breeding, to address the issue of lodging, researchers utilized CRISPR/Cas9 technology to edit a negative regulatory gene in the gibberellin signaling pathway, resulting in plants that are moderately dwarfed while maintaining high yield. This approach follows the essential logic of "clarifying gene function-selecting editing strategies-directionally regulating phenotypes" (Wang et al., 2014; Zhu et al., 2019).

2.2.4 Summary and Outlook

Gene editing technology offers a powerful approach for precisely regulating plant height. The key lies in conducting an in-depth analysis of the gene networks and signaling pathways that control height. The fundamental strategy involves selecting suitable gene-editing techniques based on the functional roles of target genes to enable directional regulation. As gene editing tools continue to evolve and research into regulatory mechanisms advances, it is anticipated that plant height can be controlled with greater precision and efficiency. This progress will provide a solid foundation for developing new crop varieties with enhanced yield potential and stress tolerance (Brophy et al., 2019; Zhao et al., 2021; Zhang et al., 2020). Nevertheless, it is equally important to address the biosafety and ethical concerns associated with gene-edited crops to ensure the responsible and sustainable application of this technology (Nicolia et al., 2014; Wolt et al., 2016).

Thank you again to the editor and reviewers for their valuable suggestions for revision. The revisions have made the language of the article more fluent, the logic clearer, and the expression more accurate.
